# YerA41, a *Yersinia ruckeri* Bacteriophage: Determination of a Non-Sequencable DNA Bacteriophage Genome via RNA-Sequencing

**DOI:** 10.3390/v12060620

**Published:** 2020-06-05

**Authors:** Katarzyna Leskinen, Maria I. Pajunen, Miguel Vincente Gomez-Raya Vilanova, Saija Kiljunen, Andrew Nelson, Darren Smith, Mikael Skurnik

**Affiliations:** 1Department of Bacteriology and Immunology, Medicum, Human Microbiome Research Program, Faculty of Medicine, University of Helsinki, 00014 UH Helsinki, Finland; katarzyna.leskinen@helsinki.fi (K.L.); maria.pajunen@helsinki.fi (M.I.P.); miguel.gomez-raya@helsinki.fi (M.V.G.-R.V.); saija.kiljunen@helsinki.fi (S.K.); 2Applied Sciences, University of Northumbria, Newcastle upon Tyne NE1 8ST, UK; andrew3.nelson@northumbria.ac.uk (A.N.); darren.smith@northumbria.ac.uk (D.S.); 3Division of Clinical Microbiology, Helsinki University Hospital, HUSLAB, 00290 Helsinki, Finland

**Keywords:** YerA41, bacteriophage, RNA-sequencing, nucleotide modification, genome assembly, transcriptome, *Yersinia ruckeri*

## Abstract

YerA41 is a *Myoviridae* bacteriophage that was originally isolated due its ability to infect *Yersinia ruckeri* bacteria, the causative agent of enteric redmouth disease of salmonid fish. Several attempts to determine its genomic DNA sequence using traditional and next generation sequencing technologies failed, indicating that the phage genome is modified in such a way that it is an unsuitable template for PCR amplification and for conventional sequencing. To determine the YerA41 genome sequence, we performed RNA-sequencing from phage-infected *Y. ruckeri* cells at different time points post-infection. The host-genome specific reads were subtracted and de novo assembly was performed on the remaining unaligned reads. This resulted in nine phage-specific scaffolds with a total length of 143 kb that shared only low level and scattered identity to known sequences deposited in DNA databases. Annotation of the sequences revealed 201 predicted genes, most of which found no homologs in the databases. Proteome studies identified altogether 63 phage particle-associated proteins. The RNA-sequencing data were used to characterize the transcriptional control of YerA41 and to investigate its impact on the bacterial gene expression. Overall, our results indicate that RNA-sequencing can be successfully used to obtain the genomic sequence of non-sequencable phages, providing simultaneous information about the phage–host interactions during the process of infection.

## 1. Introduction

YerA41 is a bacteriophage that was originally isolated due its ability to infect *Yersinia ruckeri* bacteria, the causative agent of enteric redmouth disease of salmonid fish [1]. Bacteriophage YerA41 was first described in 1984 as a tailed icosahedral virus that lysed the vast majority of tested *Y. ruckeri* serovar I strains, and thus was believed to have a potential value for the diagnosis of redmouth disease. However, it was found that YerA41 has a relatively broad host range among *Enterobacteriacae* as it could infect some strains of other *Yersinia* species, *Escherichia coli*, *Shigella flexneri*, *Enterobacter cloacae*, *Klebsiella* and *Erwinia* spp. [1]. The YerA41 phage particles are large with heads of about 110 nm in diameter, 10 × 8 nm necks, 250 × 20 nm non-contracted sheaths and 70 nm long tail fibers [2]. Several attempts to sequence the YerA41 genome using next generation sequencing (Illumina) approaches failed previously. Additionally, it was observed that the phage DNA could not be digested by any of the numerous tested restriction enzymes. These results led to the conclusion that the phage DNA is modified in such a way that prevents the function of enzymes such as DNAP and DNA-modifying restriction enzymes. Indeed, preliminary analysis of the nucleosides confirmed the presence of modifications, however, without revealing their exact nature.

Naturally occurring modifications of the canonical deoxynucleotides can be found in the DNA of organisms from all domains of life. They may constitute only a small fraction of the bases or even replace the standard, un-modified base entirely [3]. Collectively, the greatest diversity of modified bases can be observed among bacteriophages. These modifications include both the types observed among the bacterial host species, and the more unusual that are not observed among different organisms [4]. A variety of different chemical groups can be attached to the nucleotide, ranging from simple methyl groups through amino acids, polyamines, monosaccharides to more complexed oligosaccharides. They do not lead to alterations in the specificity of base pairing, and primarily appear to be a part of the arms race between the infecting organism and its host. For example, these modifications can provide additional information needed for the control of gene expression, recognition of self and non-self DNA, or protection from enzymatic degradation and host defense mechanisms [3,4].

The hypermodified bases observed among phages include 5-hydroxymethylpyrimidines and their glycosylated derivatives, α-putrescinylated and α-glutamylated thymines, sugar-substituted 5-hydroxypentyl uracil, N6-(1-acetamido)-modified adenine and 7-methylguanine [3,5]. Interestingly, the hydroxymethyl deoxypyrimidines are produced from free nucleotides by phage-encoded deoxypyrimidine hydroxymethylases before their incorporation into DNA. This mechanism is different from the primary base methylations that are commonly catalyzed in situ on DNA. After incorporation, these hydroxymethylpyrimidines often undergo further modifications introduced by phage-encoded enzymes such as glycosyl- or acetyl-transferases [3,6,7].

Here, we have used RNA-sequencing to determine the nucleotide sequence of the phage YerA41 genome using its encoded transcripts. This approach allowed us to determine the nucleotide sequence of the greater part of the phage genome and to characterize its gene expression during infection. Additionally, our approach provided insight into the interactions between YerA41 and its host bacterium. Moreover, our results indicated that YerA41 represents a novel, previously unidentified, group of bacteriophages that at the level of nucleotide sequence shares almost no similarity with those previously reported.

## 2. Materials and Methods

### 2.1. Bacterial Strains and Phage Propagation

Bacteriophage YerA41 was propagated in *Y. ruckeri* strain RS41 [1] as described previously [8]. *Y. ruckeri* was grown in lysogeny broth (LB) at room temperature (RT, 22 °C). LB agar (LA) plates were used for all solid cultures and prepared by supplementing LB with 1.5% Bacto agar and 0.4% for soft agar. Bacteriophage YerA41 was stored at −70 °C in tryptic soy broth (TSB) supplemented with 7% dimethyl sulfoxide (DMSO). The bacterial strains used in this study are listed in Appendix A.

### 2.2. Purification of Phage Particles

The standard method for the production and purification of bacteriophages [9] was used. Briefly, an overnight culture of host bacteria was diluted 10-fold in TSB in a total volume of 1 L divided into four 2 L erlenmeyer flasks, 250 mL each, and infected with appropriate number of phage to reach the multiplicity of infection (MOI) of 1. The infected cultures were incubated at 25 °C with vigorous aeration (250 rpm), until after 6–7 h the bacterial lysis took place. The lysed cultures were treated with DNase I (1.2 µg/mL; Roche Diagnostic, Mannheim, Germany) and RNase A (1 µg/mL; Sigma Chemicals, St. Louis, MO, USA) and incubated at RT for 30 min. Solid NaCl was added to a final concentration of 1 M and lysates were kept on ice for 1 h, and then the solution was centrifuged at 11,000× *g* for 10 min at 4 °C to remove the precipitated bacterial debris. The phage was recovered from the supernatant by precipitating with polyethylene glycol (PEG 8000) (10%, *w*/*v*; > 60 min; 0 °C) and was resuspended into TM buffer (50 mM Tris-HCl, pH 7.5, 10 mM MgSO_4_). Alternatively, the phage was purified from semiconfluent soft-agar plates as described [9]. The phage was further purified by chloroform extraction and one to three rounds of discontinuous glycerol density gradient ultracentrifugation at 35,000 rpm at 4 °C for 1 h in a Beckman SW41 rotor. After the centrifugations phages were resuspended in SM buffer (50 mM Tris-HCl pH 7.5, 100 mM NaCl, 8 mM MgSO_4_, 0.01% gelatin) containing 8% of sucrose.

### 2.3. One Step Growth Curve

A mid-exponential-phase culture (10 mL) of RS41 (optical density at 600 nm [OD_600_] 0.4 to 0.5) was harvested by centrifugation at 3,000 rpm for 15 min and resuspended in 1 mL of TSB medium. YerA41 phage was added at MOI of 0.0005 and allowed to adsorb for 2 min at RT. The culture was then centrifuged, the pelleted bacterial cells were resuspended to 10 mL of TSB, and incubation was continued at RT. Samples (100 µL) were taken at 10 min intervals. The first set of samples was immediately diluted and plated on lambda agar plates (per liter of broth: Bacto-agar 15 g; lambda broth per liter: tryptone 10 g, NaCl 2.5 g) for phage titration. In order to determine the eclipse period the second set of samples was treated with 1% chloroform before plating to release the intracellular phage particles [10,11]. The number of plaque forming units (PFU) in the immediately diluted 0 min time point samples was set to 1 to represent the number of infected cells in the experiment, and the PFU of all the other samples were normalized against that number. The burst size was then directly obtained from the normalized value after the rise period.

### 2.4. Total RNA Extraction

RNA extractions were carried out from three separate infection experiments. In the first experiment, the RS41 bacteria were grown for 16 h at RT and subsequently diluted 1:20 in fresh LB to a total volume of 20 mL. When the OD_600_ of the culture reached 0.7, duplicate 1 mL samples were withdrawn to represent 0 min uninfected samples, then 9 mL of the culture was infected with 1 mL of phage YerA41 stock (5 × 10^10^ pfu/mL) to achieve a MOI of ca. 50. Duplicate 1 mL aliquots were withdrawn at 5, 15, and 30 min post-infection (p.i.). In the second experiment, the RS41 bacteria were grown for 16 h at RT and subsequently diluted 1:20 in fresh LB to a total volume of 20 mL. When the OD_600_ of the culture reached 0.7, 8.5 mL of the culture was infected with 1.5 mL of phage YerA41 stock to achieve a MOI of 10. One mL aliquots were withdrawn at 33, 45, 63, 75 and 92 min p.i. in one replicate each. In the third experiment, three biological replicates were grown for 16 h at RT and diluted 1:20 in 10 mL. When the cultures reached OD_600_ of 0.6, one mL aliquots were withdrawn from each culture, these representing the uninfected 0 min samples in triplicate. Then, to 5.3 mL of culture 0.7 mL of phage stock was added to reach a MOI of 50, mixed for 3 min followed by 3 min standing and 3 min centrifugation at 4500× *g* at 22 °C. The supernatant was replaced with 6 mL fresh medium and triplicate 1 mL samples, one from each tube, were withdrawn at 15, 30 and 60 min p.i. In all experiments, total RNA was isolated from the samples using the SV Total RNA Isolation System (Promega, Madison, WI, USA) and the quality assessment was performed using LabChip (PerkinElmer, Waltham, MA, USA), using the DNA 5K/RNA/CZE chip with HT RNA Reagent Kit.

### 2.5. RNA Sequencing

The RNA-sequencing and data analysis were performed at the Nu-Omics DNA sequencing research facility at University of Northumbria. The rRNA was removed using Ribo-Zero^TM^ rRNA Removal Kit for Gram-negative Bacteria (Illumina, San Diego, CA, USA). The sequencing library was prepared using the ScriptSeq-v2 RNA-Seq Library Preparation Kit (Illumina). Paired-end sequencing was performed on MiSeq (Illumina) with the read length of 150 nucleotides.

### 2.6. Transcriptome Assembly

The obtained sequencing reads were quality filtered and aligned against *Y. ruckeri* PB-H2 chromosome (Acc.no. LN681231.1) and plasmids pYR2 (Acc.no. LN681229.1) and pYR3 (Acc.no. LN681230.1) using Bowtie2 aligner [12]. The reads that failed to align to these reference sequences were merged together and assembled using Velvet [13] and SPAdes [14]. The obtained assembled sequences were blasted against the NCBI nucleotide collection (https://blast.ncbi.nlm.nih.gov/Blast.cgi) and contigs that showed high sequence identity to *Yersinia* strains were excluded. The phage genomic scaffolds were auto-annotated using Rapid Annotation Using Subsystem Technology (RAST, http://rast.nmpdr.org/) and the obtained annotation was validated manually using the Artemis tool [15]. Presence of suitable ribosomal binding sites in front of start codons of each predicted gene was confirmed manually. The tRNA genes were identified using the ARAGORN (http://130.235.46.10/ARAGORN/) and tRNA-SCAN (http://lowelab.ucsc.edu/tRNAscan-SE/index.html) tools.

### 2.7. RNA-Sequencing Data Analysis

The quality filtered sequencing reads from different time points were aligned against the obtained YerA41 assembly scaffolds and against the *Y. ruckeri* PB-H2 reference sequence using Bowtie2 aligner [12]. The reads aligning over each gene were counted using the HTSeq [16]. The differential gene expression of bacterial transcriptome was analyzed using the edgeR [17].

### 2.8. Proteome Analysis

Purified phages were concentrated by centrifugation for 2 h at 4 °C at 16,000× *g*. Prior to the digestion of proteins to peptides with trypsin, the proteins in the samples were reduced with tris (2-carboxyethyl)phosphine (TCEP) and alkylated with iodoacetamide. Tryptic peptide digests were purified using C18 reversed-phase chromatography columns [18] and the mass spectrometry (MS) analysis was performed on an Orbitrap Elite Electron-Transfer Dissociation (ETD) mass spectrometer (Thermo Scientific, Waltham, MA, USA), using Xcalibur version 2.2, coupled to an Thermo Scientific nLC1000 nanoflow High Pressure Liquid Chromatography (HPLC) system. Peak extraction and subsequent protein identification was achieved using Proteome Discoverer 1.4 software (Thermo Scientific). Calibrated peak files were searched against the YerA41 and *Y. ruckeri* PB-H2 chromosome and plasmid proteins by a SEQUEST search engine. Error tolerances on the precursor and fragment ions were ±15 ppm and ±0.8 Da, respectively. For peptide identification, a stringent cut-off (0.05 false discovery rate or 5%) was used. The LC-MS/MS was performed at the Proteomics Unit, Institute of Biotechnology, University of Helsinki.

### 2.9. DNA Isolation

Phage DNA was obtained from high-titer phage preparations as described [9] using proteinase K plus SDS treatment followed by phenol-chloroform extractions and ethanol precipitation.

### 2.10. Accession Numbers

The RNA sequence data were deposited to the Gene Expression Omnibus (Acc. no GSE146319).

## 3. Results

### 3.1. Characterization of YerA41

The growth curve of YerA41 propagated in *Y. ruckeri* RS41 is shown in Appendix A. Apparent eclipse and latent periods of 30 and 40 min, respectively, were followed by a rise period of 20 min, and the burst size was 138 PFU per infected cell. The host range of YerA41 was studied on bacteria grown both at RT and at 37 °C, as temperature is known to regulate surface structures in the genus *Yersinia* [19]. Altogether, 129 strains representing 9 different *Yersinia* species and 3 other genera were used (Table 1 and Appendix A). All *Y. enterocolitica* strains except for those of serotype O:8 were resistant to YerA41. Most tested *Y. intermedia* and *Y. ruckeri* strains were sensitive to YerA41 phage at both tested temperatures. In addition, many *Y. kristenseni* strains, and the *Shigella flexneri* strain, were sensitive to YerA41 (Table 1). Such a broad host range indicates that the phage uses a conserved bacterial surface structure as a receptor.

### 3.2. Genome Analysis

The RNA-sequencing reads not aligned to the *Y. ruckeri* reference genome were de novo assembled; the obtained unique contigs with no sequence similarity to *Yersinia* species were scaffolded (Figure 1). This resulted in nine qualified scaffolds comprising altogether 143,296 bp (Table 2, Figure 2). The overall GC ratio of the YerA41 genome was 32.3%. Importantly, the scaffolds did not present any significant overall identity to nucleotide sequences stored in databases.

A total of 201 putative genes were detected by RAST analysis and manual annotation. Predicted functions could be assigned to 60 of the 201 gene products, the others showed no significant similarity to any protein sequences in the databases (Table 3 and Appendix A). The predicted YerA41 gene products showed similarity to DNA polymerases (Gp061, Gp097, Gp137, Gp0195) and RNA polymerase β-, β’- and other subunits (Gp019, Gp054, Gp055, Gp056, Gp162), helicases (Gp143, Gp145, Gp193), topoisomerases (Gp135, Gp136, Gp190), and a DNA ligase (Gp199). In addition, the predicted genes encode for enzymes involved in nucleoside metabolism, including 5′-deoxynucleotidase (Gp060), dCTP deaminase (Gp114), dUTP diphosphatase (Gp141), thymidylate synthetase (Gp127), ribonucleoside-diphosphate reductase (Gp110), ribonucleotide reductase (Gp109), and several endo- and exo-nucleases (Gp126, Gp149, Gp167, Gp179, Gp180, Gp189). The other gene products with recognizable function encoded different phage structural proteins. Three tRNA genes encoding tRNA-Arg, tRNA-Met, and tRNA-Leu, were identified by both ARAGORN and tRNA-SCAN, all located in a module on the scaffold_3. Finally, the gene g064-g070 products showed similarity to UDP-GlcNAc 2-epimerase, SDR family oxidoreductases, polysaccharide deacetylase, 2-C-methyl-D-erythritol-4-phosphate cytidylyltransferase, and glycerophosphodiester phosphodiesterase, suggesting them roles in biosynthesis of sugar-modified nucleotides.

### 3.3. Proteomic Analysis of the Phage Structural Proteins

To determine the phage particle associated and structural proteins, a proteomic analysis of the purified phage particles using LC-MS/MS was carried out. The structural proteins were identified through the comparative analysis of the obtained tryptic peptide sequence patterns and the sequence-based in silico determined tryptic peptide sequences of phage proteins. Altogether, 63 phage proteins were reliably identified in the LC-MS/MS analysis (Appendix A). The analysis revealed among the identified proteins structural proteins such as major capsid, tail sheath, baseplate wedge, tail fiber, and tail fiber assembly proteins. In addition, several DNA- or RNA modifying enzymes, such as DNA polymerases, helicases, recombinases, topoisomerases, endo- and exonucleases, and RNA polymerase subunits and a sigma factor were phage particle associated proteins (PPAPs, Appendix A). These are likely to be injected into the bacteria along with the genomic DNA to take over the host metabolism as soon as possible after infection (Appendix A).

### 3.4. Temporal Expression of Phage Genes

The heatmap of YerA41 genes expressed at different time points revealed evident temporal gene expression (Figure 3). The mean values of expression of early (5–15 min p.i.), middle (33–45 min), and late (63–92 min) were calculated and compared between each other. A gene was assigned to certain class based on the post infection time when its expression peaked (Table 3 and Appendix A). In total 47 genes (23.4% of phage putative genes) had their highest expression in the first min post infection. The function of the majority of the gene products remained unknown, yet six of the gene products were predicted to be involved in nucleic acid processing. Interestingly, the genes of the early phase were scattered across several scaffolds. The heatmap (Figure 3) shows that the YerA41 gene expression progresses rapidly and indistinctly to the middle phase. This phase was characterized by the expression of 101 (50.2%) genes, including numerous genes presumably involved in nucleic acid metabolism. During the latest state of infection, 51 (25.4%) genes were induced. According to the in silico analysis, the majority of these genes encode for the structural proteins of phage particles. The remaining two putative genes (the g200 and g201 genes) showed minimal expression with no changes throughout the course of infection; thus, it is highly probable that they constitute bioinformatics artifacts.

### 3.5. Bacterial Response to Lytic Infection

It is crucial to get insight into different phases of phage infection in order to understand the bacterial host response. Based on the one-step growth curve and the results of short time interval RNA-sequencing, three different time points: 15, 30, and 60 min p.i. were chosen as the representative timepoints for early, middle, and late stages of infection. Analysis of the RNA-sequencing data from three independent biological replicates revealed that *Y. ruckeri* differentially regulated the expression of 167 genes during the early stage of infection (15 min p.i.). Among the differentially expressed genes of the bacterial host (4.6%), 38% of the genes showed upregulation, and the remaining 62% showed downregulation (Table 4). Most of the upregulated genes encoded products implicated in the protection of bacterial cells from oxidative damage. These included genes encoding catalase (CSF007_12285), thioredoxin (CSF007_14055), glutaredoxin (CSF007_6685), peroxiredoxin family protein (CSF007_17480), glutathione reductase (CSF007_0665), and thioredoxin reductase (CSF007_6870). Additionally, RNA-sequencing showed the induction of genes involved in glycolysis, namely the genes encoding dihydrolipoamide dehydrogenase (CSF007_17485), glucose-6-phosphate isomerase (CSF007_16360), and pyruvate kinase (CSF007_9590), as well as three genes involved in the biosynthesis of siderophores (CSF007_15200, CSF007_15205, CSF007_15210). Among the host bacterium genes presenting the strongest upregulation p.i. was *dps* (CSF007_6300), which encodes a non-specific DNA-binding protein involved in DNA protection during exposure to severe environmental insult. An increase in expression was also observed for host bacterium genes involved in the biosynthesis of antibacterial agents like polymyxin and enterobactin (CSF007_15260, CSF007_15255, CSF007_15265). In contrast, the bacteria downregulated genes that are involved in the metabolism of different carbohydrates, as well as several genes encoding the succinate dehydrogenase complex (CSF007_5810, CSF007_5800, CSF007_5795, CSF007_5805).

At the mid time point, 30 min p.i., altogether 98 host bacterium genes presented significantly differential pattern of expression. Of the genes, 78% were significantly downregulated in response to the ongoing YerA41 infection. Further decrease in the expression of host bacterium genes encoding for succinate dehydrogenase complex (CSF007_5805, CSF007_5810, CSF007_5830, CSF007_5800, CSF007_5825, CSF007_5795) and those involved in carbohydrate metabolism and transport of sugars was observed when compared to the 15 min (early) time point (Table 4). A substantially smaller fraction (22%) of host bacterium genes were upregulated when compared to uninfected bacteria. That includes increase in expression of genes encoding cytochrome d ubiquinol oxidase subunits I and II (CSF007_5840 and CSF007_5845) and DNA-binding protein Fis (CSF007_16180)—a prominent factor implicated in bacterial gene regulation. Yet, the strongest overexpression was observed for the cold shock protein CspG (CSF007_10385). Similar to the initial phase of infection, bacteria responded by positive induction of expression of genes encoding the non-specific DNA-binding protein Dps, and catalase.

Interestingly, during the late phase, (60 min p.i.), only 15 host bacterium genes showed significantly different expression level when compared to the uninfected bacteria. All but one (phosphoenolpyruvate synthase, CSF007_9650) showed downregulation of expression. The strongest decrease in expression was observed for genes implicated in the metabolism of fructose, including the fructose-specific phosphocarrier protein HPr, fructose-specific PTS system component and 1-phosphofructokinase (CSF007_11820, CSF007_11810, CSF007_11815). Moreover, during this phase, bacteria downregulated the expression of genes encoding for factors involved in ribonucleotide reduction, such as protein NrdH and reductases of class Ib (CSF007_13980, CSF007_13985, CSF007_13990).

## 4. Discussion

In this study, we show that a transcriptomic approach can be used to obtain the genomic sequence of infectious organisms that cannot be sequenced using traditional DNA based sequencing approaches as they may possess hypermodified deoxyribonucleotides. The major advantage of the method is that it allows us to obtain the sequence of the viral transcriptome and the insight into the phage-host interaction simultaneously during the infection process. Since the phage genomes are characteristically compactly packed with minimal non-coding regions, their transcriptomes constitute the vast majority of genomic sequences, leaving very little of the unresolved genomic sequence.

We performed our RNA-sequencing study on bacterial cells infected at a relatively high MOI (MOI = 50) compared to other phage-host analyses [21,22]. At this MOI value, nearly all the bacterial cells in the culture are infected by the phage; therefore, the pattern of gene expression displayed in this study should reflect the bacterial response to lytic phage infection. One interesting phenomenon observed in this study was that the number of differentially expressed bacterial host genes decreased throughout the course of infection. We believe that it is caused by some degree of degradation of host transcripts, in combination with differences in ratio between the infected and uninfected bacteria in later phases of infection. In this situation, the degraded RNA transcripts from the infected cells would be lost.

Based upon active gene expression during infection, the YerA41 genome contains >201 putative genes; however, of these, only 60 could be assigned a predictive function. The remaining gene products exibited very limited amino acid sequence similarity to other proteins deposited in the databases, further illustrating the novelty of the YerA41 bacteriophage. Among the known gene products, there were several DNAP, RNAP β- and β’-subunits, topoisomerases, DNA ligase, helicases, as well as endo- and exonucleases. The functional analysis of these genes also revealed the presence of a putative endolysin (g49). Due to the ability to lyse bacterial cell walls, endolysins are considered to be of special interest as potential novel antimicrobials [23,24]. A very interesting group of genes were identified from scaffold 2, where the gene *g064-g070* products showed similarities sugar biosynthesis related enzymes, suggesting that they might play a role in biosynthesis of sugar-modified nucleotides.

The obtained genomic sequence indicates that YerA41 is a member of a novel, previously unidentified, group of bacteriophages. At the nucleotide level it shares no significant similarity with genomic sequences of any known organisms deposited in the databases. Conversely, the in silico analysis revealed the presence of multiple unique proteins that are predicted to be involved in nucleic acid processing and metabolism. Taking this into consideration, it is logical that it is equipped with its own machinery for transcription and amplification of the genetic information. At the moment, the exact nature of the nucleotide modification present in the genome of YerA41 is still unknown, but the research tackling this question is ongoing.

## Figures and Tables

**Figure 1 viruses-12-00620-f001:**
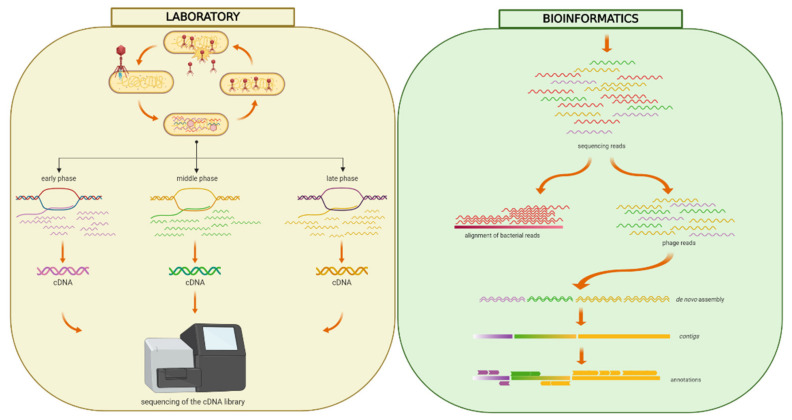
Workflow of the RNA-seq approach. Freshly diluted *Y. ruckeri* RS41 bacteria were grown until OD_600_ = 0.6 and then infected with phage YerA41 at MOI equal to 10 or 50. The culture was washed with LB to remove the unbound phage particles, and thus prevent re-infection of bacterial cells at later stages of the experiment. Samples for RNA isolation were taken at different time points p.i. (0, 5, 15, 33, 45, 63, 75, 92 min). After the removal of bacterial rRNA, the prepared libraries were sequenced. The obtained sequencing reads were quality filtered and aligned against *Y. ruckeri* PBH2 chromosome (Acc.no. LN681231.1) and plasmids pYR2 (Acc.no. LN681229.1) and pYR3 (Acc.no. LN681230.1). The reads that failed to align to these reference sequences were merged together and assembled using Velvet [13] and SPAdes [14]. The obtained assembled sequences were blasted against the NCBI nucleotide collection and contigs showing high identity rates with *Yersinia* strains were excluded. The phage genomic scaffolds were auto-annotated using Rapid Annotation Using Subsystem Technology (RAST) [20]. Presence of suitable ribosomal binding sites in front of each predicted start codon was confirmed. Figure was created using BioRender (https://app.biorender.com).

**Figure 2 viruses-12-00620-f002:**
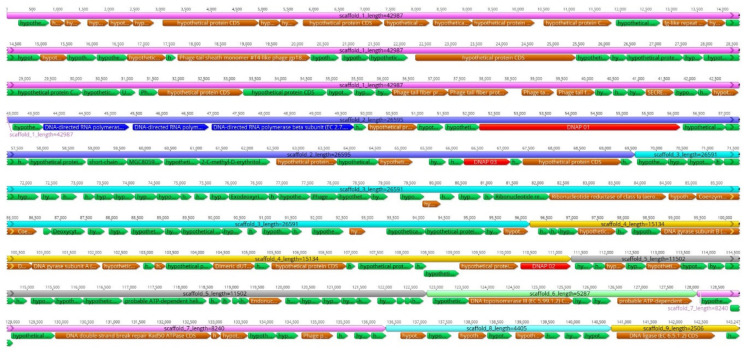
Gene organization of the nine scaffolds of the phage YerA41 genome. The scaffolds are organized based on size. The gene colors indicate predicted functions of their products: Green, hypothetical proteins; Blue, RNA polymerase subunits; Red, DNA polymerase-like proteins; Brown, phage particle associated (structural) proteins. The figure was generated using Geneious 10.2.6 (www.geneious.com).

**Figure 3 viruses-12-00620-f003:**
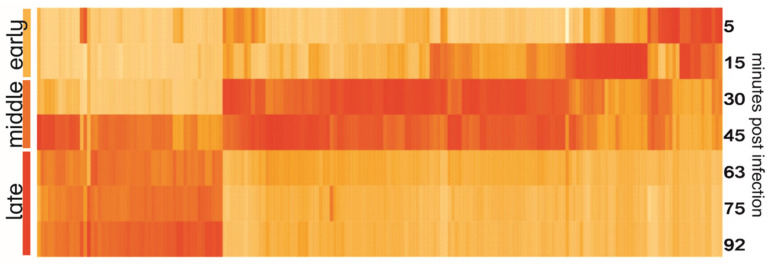
Heatmap of YerA41 genes expressed at different time points post infection. Gene expression values were normalized to the highest expression to show the timing of expression; therefore, the intensity of color on the heatmap reflects the difference of expression of one gene at different time points, yet not the difference of expression between different genes.

**Table 1 viruses-12-00620-t001:** Host range of bacteriophage YerA41 (numbers of studied strains for each species and serotype are given in parenthesis). The sensitivity was tested both at RT and at 37 °C.

Bacterial Species	YerA41 Sensitive Serotypes	YerA41 Sensitive Serotypes at 37 °C	YerA41 Resistant Serotypes or Strains
*Yersinia enterocolitica*	O:8 (5)	O:8 (5)	O:1(2),O:1,2,3(1), O:2(2), O:3(2), O:4(1), O:4,32(2), O:5(3), O:5,27(2), O:6(2), O:6,30(2), O:6,31(2), O:7,8(8), O:9(2), O:13,17(1), O:13,7(2), O:13a,13b(2), O:14(1), O:15(2) O:20(2), O:21(2), O:25(1), O:25,26(1), O:26,44(1), O:28,50(1), O:34 (1), O:35,36 (1), O:35,52(1), O:41(27)43(2), O:41(27)42 K1(1), O:50(2), O:41(27)K1(1),O:41,43(1), K1 non-typable (2), non-typable(4)
*Yersinia pseudo-tuberculosis*	-	-	O:1(1), O:1a(1), O:1b(1), O:2 (2), O:2a(1), O:2b(1), O:2c(1), O:3 (2), O:4a (1), O:4b (1), O:5a (1), O:5b(1), O:6 (1), O:7 (1), O:8 (1), O:9(1),
*Yersinia frederikseni*	-	O:16(1),O:35(1)	O:48 (1), non-typable (4)
*Yersinia intermedia*	O:16,21(1), O:52,54 (1)	O:16,21(1)	-
*Yersinia kristenseni*	O:12,25(1), O:16(2), non-typable (1)	O:12.25(1), O:16(2), non-typable (1)	non-typable(1)
*Yersinia mollareti*	-	-	O:59(20,36,7) (1)
*Yersinia pestis*	-	-	(2)
*Yersinia bercoveri*	-	-	O:58,16(1), non-typable(1)
*Yersinia ruckeri*	(2)	(2)	-
*Providencia rettgeri*	-	-	(1)
*Salmonella typhimurium*	-	-	(1)
*Shigella flexneri*	(1)	(1)	-

**Table 2 viruses-12-00620-t002:** The assembled scaffolds of YerA41 genome. The numbering of the scaffolds is based on their length and does not reflect their actual position in the phage genome.

ID	Size [bp]	GC%
scaffold_1	42 987	34.5
scaffold_2	27 377	31.6
scaffold_3	26 591	31.7
scaffold_4	15 134	32.0
scaffold_5	11 502	32.8
scaffold_6	8 240	29.6
scaffold_7	5 287	28.9
scaffold_8	3 672	29.6
scaffold_9	2 506	29.6
**TOTAL/Average:**	**143 296**	**32.3**

**Table 3 viruses-12-00620-t003:** Temporal expression profiles of the gene products of phage YerA41 having predicted functions. The full list of all gene products, including the hypothetical proteins of unknown function, is presented in Appendix A. The LC-MS/MS-identified PPAPs are marked with asterisk.

**Temporal Expression**	**Gene ID**	**Scaffold**	**Putative Functions of Gene Products Based on Database Similarity**
Early	*g054*	2	DNA directed RNA polymerase, subunit*
	*g055*	2	DNA-directed RNA polymerase, subunit*
	*g056*	2	DNA-directed RNA polymerase*
	*g078*	2	Lytic transglycosylase
	*g097*	3	DNA polymerase III, subunit
	*g116*	3	RNA 2’-phosphotransferase
	*g126*	3	Endonuclease-like protein
	*g135*	4	DNA topoisomerase*
	*g161*	5	Tail fiber protein
	*g162*	5	DNA directed RNA polymerase, subunit / Putative DNA helicase
	*g189*	7	Endonuclease-like protein
	*g190*	7	DNA topoisomerase*
	*g193*	7	Helicase*
	*g195*	8	DNA polymerase
	*g199*	9	DNA ligase*
Middle	*g060*	2	5’-deoxynucleotidase
	*g061*	2	DNA polymerase*
	*g064*	2	UDP-GlcNAc 2-epimerase
	*g065*	2	Oxidoreductase
	*g066*	2	SDR family oxidoreductase
	*g067*	2	Polysaccharide deacetylase
	*g068*	2	2-C-methyl-D-erythritol 4-phosphate cytidylyltransferase (EC 2.7.7.60)
	*g070*	2	Glycerophosphodiester phosphodiesterase
	*g109*	3	Ribonucleotide reductase
	*g110*	3	Ribonucleoside-diphosphate reductase subunit alpha*
	*g112*	3	Ribosomal protein modification protein*
	*g114*	3	dCTP deaminase
	*g127*	3	Thymidylate synthetase
	*g133*	4	Transglycosylase
	*g136*	4	DNA topoisomerase*
	*g137*	4	DNA polymerase III, subunit*
	*g140*	4	Structural protein
	*g141*	4	dUTP diphosphatase*
	*g143*	4	ATP-dependent DNA helicase*
	*g145*	4	Replicative helicase
	*g149*	4	Exodeoxyribonuclease*
	*g167*	5	Endonuclease*
	*g174*	5	Phage baseplate assembly protein
	*g179*	6	Exonuclease
	*g180*	6	Exonuclease*
Late	*g007*	1	DNA packaging terminase*
	*g012*	1	Prohead core protein protease*
	*g014*	1	Capsid protein*
	*g016*	1	Sugar binding protein*
	*g019*	1	RNA polymerase sigma factor*
	*g024*	1	Phage tail sheath protein*
	*g025*	1	Tail protein
	*g028*	1	Tail family protein*
	*g031*	1	Tail protein
	*g032*	1	Tail protein
	*g034*	1	Tail-associated lysozyme
	*g035*	1	Tail-associated lysozyme
	*g037*	1	Baseplate wedge protein*
	*g040*	1	Capsid protein
	*g041*	1	Virion structural protein
	*g042*	1	Baseplate wedge protein*
	*g043*	1	Tail fiber protein*
	*g044*	1	Phage tail fiber assembly protein*
	*g045*	1	Tail fiber protein*
	*g049*	1	Endolysin*

**Table 4 viruses-12-00620-t004:** Transcriptional response of *Y. ruckeri* to infection with YerA41. The list of bacterial genes showing significant (*p*-value < 0.001) differential expression at both 15 min and 30 min time points compared to non-infected bacteria. The lists of genes differentially expressed at different time points (15, 30 and 60 min p.i.) are presented in Appendix A. LogFC; log-ratio of a transcript’s expression values in two different conditions. FDR; False Discovery Rate.

**Gene ID**	**Function**	**15 min**		**30 min**	
		**logFC**	**FDR**	**logFC**	**FDR**
CSF007_17485	Dihydrolipoamide dehydrogenase	7.07	1.97 × 10^−146^	1.28	2.16 × 10^−07^
CSF007_17480	Peroxiredoxin family protein/glutaredoxin	6.95	4.94 × 10^−128^	1.12	7.36 × 10^−05^
CSF007_6300	Non-specific DNA-binding protein Dps / Iron-binding ferritin-like antioxidant protein / Ferroxidase	4.39	4.13 × 10^−66^	1.90	5.48 × 10^−11^
CSF007_12285	Catalase	4.10	3.26 × 10^−59^	1.54	9.40 × 10^−09^
CSF007_9590	Pyruvate kinase	1.95	7.27 × 10^−15^	2.11	9.00 × 10^−13^
CSF007_11760	Putrescine importer	1.43	1.44 × 10^−07^	1.71	4.96 × 10^−05^
CSF007_5840	Cytochrome d ubiquinol oxidase subunit I	1.21	1.89 × 10^−06^	1.49	0.00011
CSF007_5845	Cytochrome d ubiquinol oxidase subunit II	1.14	1.98 × 10^−06^	1.48	2.78 × 10^−05^
CSF007_13405	Inosine-5-monophosphate dehydrogenase	0.91	0.00034	1.19	0.00017
CSF007_12920	hypothetical protein	0.88	0.00014	1.11	0.00015
CSF007_5505	hypothetical protein	−0.78	1.77 × 10^−05^	−1.18	1.44 × 10^−05^
CSF007_14715	Glycine cleavage system H protein	−0.81	0.00044	−1.88	3.38 × 10^−07^
CSF007_9025	Alkyl sulfatase	−0.93	2.94 × 10^−06^	−1.32	1.03 × 10^−05^
CSF007_13885	D-ribulokinase	−0.94	6.94 × 10^−07^	−2.04	9.33 × 10^−14^
CSF007_13880	Phosphosugar isomerase/binding protein	−1.01	2.19 × 10^−06^	−1.98	6.52 × 10^−09^
CSF007_1760	Aspartate ammonia-lyase	−1.02	6.88 × 10^−05^	−1.98	3.86 × 10^−12^
CSF007_0675	Oligopeptidase A	−1.03	0.00043	−1.29	0.00043
CSF007_9680	Hemin transport protein HmuS	−1.06	1.77 × 10^−05^	−1.38	0.00097
CSF007_17975	Glutamine synthetase type I	−1.09	0.00014	1.92	4.07 × 10^−05^
CSF007_14720	Aminomethyltransferase (glycine cleavage system T protein)	−1.11	3.59 × 10^−09^	−1.69	7.18 × 10^−10^
CSF007_11035	Transcriptional repressor of PutA and PutP / Proline dehydrogenase (Proline oxidase) / Delta-1-pyrroline-5-carboxylate dehydrogenase	−1.12	9.24 × 10^−06^	−2.14	5.30 × 10^−14^
CSF007_13080	NADP-dependent malic enzyme	−1.12	9.98 × 10^−08^	−1.68	2.82 × 10^−09^
CSF007_6400	Galactose/methyl galactoside ABC transport system ATP-binding protein MglA	−1.13	1.92 × 10^−07^	−1.72	6.99 × 10^−09^
CSF007_0690	Universal stress protein A	−1.20	1.55 × 10^−07^	−1.56	5.98 × 10^−06^
CSF007_0605	Aerobic C4-dicarboxylate transporter for fumarate/L-malate/D-malate/succunate	−1.23	1.07 × 10^−09^	−1.09	0.00046
CSF007_1210	Cyclic AMP receptor protein	−1.32	5.98 × 10^−08^	−1.42	1.30 × 10^−05^
CSF007_0245	16 kDa heat shock protein A	−1.37	0.00033	−1.80	6.55 × 10^−06^
CSF007_5820	Dihydrolipoamide succinyltransferase component (E2) of 2-oxoglutarate dehydrogenase complex	−1.41	7.97 × 10^−08^	−2.87	1.10 × 10^−12^
CSF007_18075	Ribose ABC transport system periplasmic ribose-binding protein RbsB	−1.41	1.96 × 10^−11^	−1.57	8.42 × 10^−07^
CSF007_16000	hypothetical protein	−1.42	2.65 × 10^−06^	−1.99	6.12 × 10^−06^
CSF007_11865	Mannonate dehydratase	−1.46	8.78 × 10^−10^	−2.13	6.04 × 10^−12^
CSF007_13895	Ribose ABC transport system permease protein RbsC	−1.48	1.29 × 10^−12^	−2.07	7.16 × 10^−12^
CSF007_0935	Transcriptional activator of maltose regulon MalT	−1.49	7.07 × 10^−14^	−1.48	8.42 × 10^−07^
CSF007_16315	Maltose operon periplasmic protein MalM	−1.51	6.85 × 10^-06^	−2.03	0.00043
CSF007_18085	Ribose ABC transport system ATP-binding protein RbsA	−1.51	7.72 × 10^−10^	−1.85	2.56 × 10^−06^
CSF007_9675	TonB-dependent hemin ferrichrome receptor	−1.55	9.46 × 10^−16^	−1.13	0.00018
CSF007_5825	Succinyl-CoA ligase [ADP-forming] beta chain	−1.60	7.19 × 10^−08^	−2.86	5.80 × 10^−10^
CSF007_5830	Succinyl-CoA ligase [ADP-forming] alpha chain	−1.63	1.56 × 10^−09^	−3.01	4.78 × 10^−14^
CSF007_18090	Ribose ABC transport system high affinity permease RbsD	−1.66	1.10 × 10^−11^	−2.16	4.82 × 10^−08^
CSF007_16340	Maltose/maltodextrin ABC transporter substrate binding periplasmic protein MalE	−1.68	5.91 × 10^−08^	−2.03	1.45 × 10^−06^
CSF007_3355	Aconitate hydratase 2	−1.69	3.40 × 10^−12^	−1.66	1.56 × 10^−06^
CSF007_5815	2-oxoglutarate dehydrogenase E1 component	−1.80	2.19 × 10^−13^	−2.96	3.74 × 10^−18^
CSF007_9550	Putative transport protein	−1.80	4.34 × 10^−12^	−1.51	2.19 × 10^−06^
CSF007_16325	Maltose/maltodextrin transport ATP-binding protein MalK	−1.81	6.80 × 10^−06^	−2.93	3.65 × 10^−08^
CSF007_9650	Phosphoenolpyruvate synthase	−1.82	6.56 × 10^−15^	−2.54	1.16 × 10^−09^
CSF007_11875	D-mannonate oxidoreductase	−1.87	3.62 × 10^−13^	−2.57	4.08 × 10^−14^
CSF007_12965	Sialic acid transporter (permease) NanT	−1.92	2.90 × 10^−12^	−2.72	4.08 × 10^−14^
CSF007_13900	Ribose/xylose/arabinose/galactoside ABC-type transport system ATP-binding protein	−2.08	2.79 × 10^−28^	−2.06	7.97 × 10^−05^
CSF007_6395	Galactose/methyl galactoside ABC transport system galactose-binding periplasmic protein MglB	−2.08	2.72 × 10^−14^	−2.79	1.40 × 10^−17^
CSF007_11455	hypothetical protein	−2.12	8.23 × 10^−28^	−1.57	4.48 × 10^−05^
CSF007_0865	Gluconokinase	−2.12	9.37 × 10^−17^	−1.80	7.28 × 10^−06^
CSF007_12460	membrane protein	−2.21	3.34 × 10^−18^	−2.35	7.75 × 10^−09^
CSF007_15720	Hexuronate transporter	−2.38	7.15 × 10^−28^	−2.40	1.25 × 10^−10^
CSF007_17650	Glycerol uptake facilitator protein	−2.39	1.55 × 10^−28^	−2.08	5.48 × 10^−11^
CSF007_16005	Trehalose-6-phosphate hydrolase	−2.46	3.37 × 10^−14^	−3.86	3.31 × 10^−24^
CSF007_5810	Succinate dehydrogenase iron-sulfur protein	−2.47	1.33 × 10^−16^	−3.06	6.67 × 10^−15^
CSF007_13910	Ribose/xylose/arabinose/galactoside ABC-type transport system periplasmic sugar binding protein	−2.48	1.26 × 10^−19^	−3.70	1.55 × 10^−33^
CSF007_5800	Succinate dehydrogenase hydrophobic membrane anchor protein	−2.49	4.39 × 10^−19^	−2.64	1.15 × 10^−10^
CSF007_17655	Glycerol kinase	−2.56	1.20 × 10^−19^	−3.31	2.81 × 10^−15^
CSF007_5795	Succinate dehydrogenase cytochrome b-556 subunit	−2.64	4.76 × 10^−26^	−2.04	9.68 × 10^−13^
CSF007_15715	hypothetical protein	−2.64	2.61 × 10^−10^	−3.74	1.21 × 10^−17^
CSF007_5805	Succinate dehydrogenase flavoprotein subunit	−2.65	2.01 × 10^−22^	−3.18	3.94 × 10^−18^
CSF007_12450	Ascorbate-specific PTS system, EIIA component	−3.07	6.74 × 10^−23^	−3.17	6.76 × 10^−10^
CSF007_12455	Putative sugar phosphotransferase component II B	−3.21	1.62 × 10^−22^	−3.52	1.76 × 10^−09^
CSF007_5790	Citrate synthase	−3.23	8.51 × 10^−22^	−3.12	8.36 × 10^−10^
CSF007_13905	hypothetical protein	−3.28	2.14 × 10^−13^	−5.24	6.87 × 10^−10^
CSF007_16010	PTS system, trehalose-specific IIB component-PTS system	−3.39	3.53 × 10^−41^	−3.85	1.29 × 10^−27^

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
