# Peer review of "YerA41, a Yersinia ruckeri Bacteriophage: Determination of a Non-Sequencable DNA Bacteriophage Genome via RNA-Sequencing"

_viruses, 2020, doi:10.3390/v12060620_

Round 1

Reviewer 1 Report

The paper describes a new methods of sequencing bacteriophage through transcriptomics. The approach is interesting and worthwhile since many bacteriophage have modified DNA that is not conducive to sequencing and this approach has the benefit of giving gene expression data for both the bacteriophage and it’s host, as well as the genome sequence of the phage. However, while the paper is interesting and the topic is worth publishing, as written the paper is not of a quality that it can be published in its current form. In particular there are no statistics for the results in figures 1 and 2.  Also, the discussion is short and doesn’t really discuss the results of the paper in any depth.

Minor comments

Line 45 - Does DNAP refer to DNA polymerase? If so I don’t think should be considered a DNA modifying enzyme.

Figure 1,2 – error bars.

Line 185 and throughout – Species and genus names should be italicized

Figure 3 - I don’t think this figure is necessary, also it would belong in the materials and methods section

Line 262 – I believe you mean Figure 5

Figure 5 – This figure is very confusing, only scaffold 1 is marked on the x-axis. It also isn’t clear what is happening in this figure, there are the three conditions but early and middle have two rows of results and late has three rows of results.

Author Response

Reviewer #1

The paper describes a new methods of sequencing bacteriophage through transcriptomics. The approach is interesting and worthwhile since many bacteriophage have modified DNA that is not conducive to sequencing and this approach has the benefit of giving gene expression data for both the bacteriophage and it’s host, as well as the genome sequence of the phage. However, while the paper is interesting and the topic is worth publishing, as written the paper is not of a quality that it can be published in its current form. In particular there are no statistics for the results in figures 1 and 2.  Also, the discussion is short and doesn’t really discuss the results of the paper in any depth.

REPLY:

Minor comments:

Line 45 - Does DNAP refer to DNA polymerase? If so I don’t think should be considered a DNA modifying enzyme.

REPLY: The text has been modified as suggested

Figure 1,2 – error bars.

REPLY: We have faced severe difficultied in repeating the experiments due to the corona crisis. Therefore we decided to move Figure 1 to Supplementary data and delete figure 2. Both of the experiments are not central to the manuscript therefore this decision.

Line 185 and throughout – Species and genus names should be italicized

REPLY: Species and genus names were italicized throughout the document.

Figure 3 - I don’t think this figure is necessary, also it would belong in the materials and methods section

REPLY: In our opinion, this figure helps the readers to understand the flow of the experimentation procedure. It summarizes well the whole procedure we presented in this publication.

Line 262 – I believe you mean Figure 5

REPLY: The number of the figure marked in the brackets was corrected.

Figure 5 – This figure is very confusing, only scaffold 1 is marked on the x-axis. It also isn’t clear what is happening in this figure, there are the three conditions but early and middle have two rows of results and late has three rows of results.

REPLY: The figure was modified. The marking of the scaffold 1 was removed and each sample was marked according to its time of isolation. Additionally, the legend of the figure was modified to make it easier to understand.

Reviewer 2 Report

The authors carried out a novel and interesting experiment to produce an approximation of the Yersinia phage YerA41 genome, which was originally isolated almost 40 years ago but has not been successfully sequenced. To overcome what the authors posit is a problem of phage DNA modification, they produced multiple phage cDNA libraries which they assembled into a model genome.

On the whole, this is an interesting experiment done well that requires only minor revisions. 

Line 41- I would be curious to know which next generation sequencing methods the authors tried. Some, such as SMRT are better at dealing with, and identifying, modified bases than others.

Lines 174 - 193, Bacterial genus and species should be italicized.

Fig 1. I would consider putting a horizontal line beneath the vertical arrow suggesting how the burst size was determined. The lower end point of the arrow is open to interpretation. It would also be reasonable to provide more detail on how you calculated the burst size in the one step growth curve section of the Materials and Methods.

Fig 2. The authors indicate that each data point is the mean of three experiments. I think error bars might be appropriate and helpful. 

Line 262 - do you mean fig 5?

Fig 5. The portion of the data from scaffold 1 is highlighted though it is not immediately clear why it is noted but no others are.

Line 307-326 - The authors discussed gene expression levels in both the phage and in the host  in the previous paragraph, then continues the topic in these paragraphs. It is not immediately obvious which is being referred to and that the authors have switched back to the phage. 

Lines 328-329, 343-345 - The introduction suggests that DNA modification is the reason for the difficulty in sequencing (line 42-46), and that may be, but no evidence of is presented in this manuscript or referenced from another. To state in the discussion that the DNA is modified without evidence seems unwarranted. 

Line 348 - I am very impressed and excited by this protocol. But of course it has the limitation of only seeing active genes. It is just not possible to know how many phage genes were invisible because they were inactive or at an undetectable level of expression. The very reason a final genome couldn’t be assembled is the reason a final phage gene total can’t be known, but rather an estimate based on active genes (as indicated in line 69-70).

The is some minor editing needed for readability, for example: 

  • Line 13 “Myoviridae
  • Line 54-56, this is an awkward sentence
  • Line 62, this sentence doesn’t make sense starting with “opposed to” 
  • Line 87, “were allowed to cool to RT, then treated,” unless they were cooled?
  • Line 97, “containing 8% sucrose”
  • Line 99, “3,000”
  • Line 112-113, this is an awkward sentence
  • Line 286, “of” 
  • Line 295, “a gene that encodes a” or “which encodes a”
  • Lines 298-300, this sentence is not clear, it could be fixed by changing it to “that play roles” or “that play roles,” and “In contrast” makes more sense than on the contrary. But you might consider rewriting this.
  • Lines 295 and 318, DPS is all lowercase in one place and capitalized in the other, should standardize
  • Line 354, also revealed 

Author Response

Reviewer #2

The authors carried out a novel and interesting experiment to produce an approximation of the Yersinia phage YerA41 genome, which was originally isolated almost 40 years ago but has not been successfully sequenced. To overcome what the authors posit is a problem of phage DNA modification, they produced multiple phage cDNA libraries which they assembled into a model genome.

On the whole, this is an interesting experiment done well that requires only minor revisions. 

Line 41- I would be curious to know which next generation sequencing methods the authors tried. Some, such as SMRT are better at dealing with, and identifying, modified bases than others.

REPLY: We used Illumina NGS. PacBio and Nanopore approaches were not tried. This is now stated in the text.

Lines 174 - 193, Bacterial genus and species should be italicized.

REPLY: Species and genus names were italicized throughout the document.

Fig 1. I would consider putting a horizontal line beneath the vertical arrow suggesting how the burst size was determined. The lower end point of the arrow is open to interpretation. It would also be reasonable to provide more detail on how you calculated the burst size in the one step growth curve section of the Materials and Methods.

REPLY: We have revised the text in the methods to explain the burst size calculation and moved the figure to supplementary information as we were not able to repeat the experiments properly due to the corona epidemics.

Fig 2. The authors indicate that each data point is the mean of three experiments. I think error bars might be appropriate and helpful. 

REPLY: We have removed the figure from the manuscript as we realized that it was lacking a critical control for YerA41, the adsorption values for non-heated bacteria. As the figure is not central issue in this manuscript we decided to delete it.

Line 262 - do you mean fig 5?

REPLY: The number of the figure marked in the brackets was corrected.

Fig 5. The portion of the data from scaffold 1 is highlighted though it is not immediately clear why it is noted but no others are.

REPLY: The figure was modified. The marking of the scaffold 1 was removed and each sample was marked according to its time of isolation. Additionally, the legend of the figure was modified to make it easier to understand.

Line 307-326 - The authors discussed gene expression levels in both the phage and in the host  in the previous paragraph, then continues the topic in these paragraphs. It is not immediately obvious which is being referred to and that the authors have switched back to the phage. 

REPLY: All the text discusses the host genes and we have revised the text and now it is clearly indicated

Lines 328-329, 343-345 - The introduction suggests that DNA modification is the reason for the difficulty in sequencing (line 42-46), and that may be, but no evidence of is presented in this manuscript or referenced from another. To state in the discussion that the DNA is modified without evidence seems unwarranted. 

REPLY: As suggested we have revised the text to address this criticism.

Line 348 - I am very impressed and excited by this protocol. But of course it has the limitation of only seeing active genes. It is just not possible to know how many phage genes were invisible because they were inactive or at an undetectable level of expression. The very reason a final genome couldn’t be assembled is the reason a final phage gene total can’t be known, but rather an estimate based on active genes (as indicated in line 69-70).

 REPLY: This is correct and of course the obvious limitation, however, it proved to be the only possibility to obtain any sequence from this phage by the approaches we had used that far.

The is some minor editing needed for readability, for example: 

  • Line 13 “Myoviridae”
  • Line 54-56, this is an awkward sentence
  • Line 62, this sentence doesn’t make sense starting with “opposed to” 
  • Line 87, “were allowed to cool to RT, then treated,” unless they were cooled?
  • Line 97, “containing 8% sucrose”
  • Line 99, “3,000”
  • Line 112-113, this is an awkward sentence
  • Line 286, “of” 
  • Line 295, “a gene that encodes a” or “which encodes a”
  • Lines 298-300, this sentence is not clear, it could be fixed by changing it to “that play roles” or “that play roles,” and “In contrast” makes more sense than on the contrary. But you might consider rewriting this.
  • Lines 295 and 318, DPS is all lowercase in one place and capitalized in the other, should standardize. line 295 means the gene, now in italics, and line 318, the protein
  • Line 354, also revealed.

REPLY: All revised as suggested

Round 2

Reviewer 1 Report

I am satisfied that all my concerns have been addressed.